# Tumor Neurobiology in the Pathogenesis and Therapy of Head and Neck Cancer

**DOI:** 10.3390/cells13030256

**Published:** 2024-01-30

**Authors:** Siyuan Liang, Jochen Hess

**Affiliations:** 1Department of Otorhinolaryngology, Head and Neck Tumors, Heidelberg University Hospital, 69120 Heidelberg, Germany; siyuan.liang@stud.uni-heidelberg.de; 2Research Group Molecular Mechanisms of Head and Neck Tumors, German Cancer Research Center (DKFZ), 69120 Heidelberg, Germany

**Keywords:** head and neck cancer, nerve–cancer crosstalk, TME, tumor neurobiology

## Abstract

The neurobiology of tumors has attracted considerable interest from clinicians and scientists and has become a multidisciplinary area of research. Neural components not only interact with tumor cells but also influence other elements within the TME, such as immune cells and vascular components, forming a polygonal relationship to synergistically facilitate tumor growth and progression. This review comprehensively summarizes the current state of the knowledge on nerve–tumor crosstalk in head and neck cancer and discusses the potential underlying mechanisms. Several mechanisms facilitating nerve–tumor crosstalk are covered, such as perineural invasion, axonogenesis, neurogenesis, neural reprogramming, and transdifferentiation, and the reciprocal interactions between the nervous and immune systems in the TME are also discussed in this review. Further understanding of the nerve–tumor crosstalk in the TME of head and neck cancer may provide new nerve-targeted treatment options and help improve clinical outcomes for patients.

## 1. Introduction

### 1.1. Nervous System (CNS/PNS)

The nervous system, consisting of the central nervous system (CNS) and peripheral nervous system (PNS), controls a wide range of physiological activities in life, such as organogenesis and development, homeostasis, and regeneration, and modulates many pathophysiological processes, including cancer [1,2]. The CNS, which consists of the brain, brainstem, cerebellum, and spinal cord, regulates tumorigenesis and growth by releasing neurotransmitters or hormones into the systemic circulation [1,3]. In addition, the CNS also functions through the PNS, which acts as a bridge to establish connections between the CNS and the local tumor tissue. The PNS, composed of motor, sensory, and autonomic (sympathetic and parasympathetic) nerve fibers, branches throughout the body, usually accompanied by the microvasculature, infiltrating the tumor microenvironment to modulate vascular or cellular elements [1]. A growing body of experimental and clinical evidence indicates that sympathetic and parasympathetic nerves modulate tumor progression in an antagonistic manner. For instance, sympathetic nerves (adrenergic signaling) can accelerate tumor growth, whereas parasympathetic nerves (cholinergic signaling) have the opposite effect in pancreatic and breast cancer [4,5,6]. However, the effect may be reversed depending on the type of cancer, as cholinergic signaling has been reported to promote tumor growth in gastric cancer [7,8]. In addition, parasympathetic innervation has been found to be critical for glandular tubulogenesis, organogenesis, and regeneration in the salivary gland, demonstrating its pivotal role in stem cell maintenance [9,10]. The influence of sensory nerves on various cancers has also been reported [11,12,13]. Reciprocally, tumor cells can reversely remodel the nervous system via the direct contact or release of neurotrophic growth factors or chemokines [14,15].

### 1.2. Tumor Microenvironment

The tumor microenvironment (TME) is an ecosystem composed of various cell types, such as endothelial cells, fibroblasts, immune cells, and extracellular matrix components [16,17]. The nervous system has emerged as a new pathological component of the TME and, together with other cellular and non-cellular components, forms a neural–immune–vascular network that plays an important role in tumor initiation, progression, and metastasis. The complex crosstalk between tumor cells and the surrounding environment has been widely recognized to influence therapeutic efficacy [18,19]. 

The diameters and densities of nerves and their distances from cancer cells are critical features for describing the neurophenotype within the TME and have a clinically relevant impact on the nerve–tumor crosstalk [20]. The nerve diameter is the most fundamental element, and several studies have reported that larger nerve diameters are associated with poor clinical outcomes in several cancer types, including pancreatic, gastric, and head and neck cancers [21,22,23]. The nerve density, defined as the number of nerves in a given area and mainly influenced by axonogenesis and neurogenesis, is another significant predictor of an unfavorable prognosis [24,25,26]. The distance between nerves and cancer cells in the TME, which varies from distant to close but without physical contact to perineural invasion (PNI), is another critical parameter related to tumor metastasis and the death rate [23,27].

As an important component of the TME, the nervous system regulates pathological processes by secreting neurotransmitters and growth factors to enhance tumor development. Conversely, tumor cells remodel the nervous system and alter neurological functions to support their growth and promote intra-tumoral innervation. Nerves interact with other components of the TME to provide metabolic support for tumor cells. Neural signals can regulate and coordinate vascular growth to establish a vascular network to ensure the transport of oxygen and nutrients [28]. In prostate cancer, noradrenaline secreted by adrenergic nerves can activate angiogenesis via endothelial β-adrenergic receptor signaling to facilitate tumor growth. In a mouse model, the loss of endothelial *Adrb2*, which encodes the β2-adrenergic receptor, altered the endothelial cell metabolism from aerobic glycolysis to oxidative phosphorylation, leading to the suppression of angiogenesis [29]. In pancreatic ductal adenocarcinoma, the peripheral axons supply serine to provide metabolic support for tumor growth in nutrient-poor environments, while serine-deprived conditions can conversely promote tumor innervation via the translation and secretion of the nerve growth factor (NGF) [30]. Tumor cells have also been reported to exploit nociceptive nerves to produce calcitonin gene-related peptide (CGRP), thereby inducing cytoprotective autophagy to thrive in nutrient-poor environments [11]. 

### 1.3. Neurobiology of Head and Neck Tumors

Head and neck cancer, which mainly includes malignancies of the oral and maxillofacial regions and upper aerodigestive tract, is the seventh most common cancer type worldwide, with high morbidity and mortality rates [31,32]. Tobacco use, alcohol consumption, human papillomavirus (HPV) infection, and Epstein–Barr virus (EBV) infection are widely recognized risk factors for head and neck cancer [32,33], and approximately 90% are diagnosed as squamous cell carcinoma (HNSCC) [34]. Traditional treatment options include surgery, chemotherapy, and radiotherapy as monotherapy or in combination, depending on the tumor location and staging. In addition, risk factors such as the resection margin, extracapsular spread, and HPV status also influence postoperative treatment strategies [32,35]. Emerging, new treatment strategies, such as immunotherapy and targeted therapy, demonstrate the increasing focus on key elements of the TME [32,36]. These elements of the TME include neural components, which are common because head and neck cancers, especially oral cancer, exhibit perineural invasion and intra-tumoral innervation to a greater extent than other cancers due to the highly innervated nature of the head and neck region. Recent review articles have summarized biomarkers and neural markers involved in perineural invasion, axonogenesis, and neural reprogramming (discussed in the following chapters), providing compelling evidence for nerve–tumor crosstalk in head and neck cancer [37,38]. Several preclinical studies have shown that the surgical or chemical ablation of nerve fibers can significantly affect the tumor growth in prostate cancer, gastric cancer, and pancreatic cancer [8,12,28]. Due to the important functions of nerves in the facial region, the application of ablation therapy in head and neck cancer is relatively limited. In a mouse model of orthotopically injected p53-deficient oral squamous cell carcinoma (OSCC) cells, which cause tumor-associated sensory nerves to transdifferentiate into adrenergic nerves, treatment with carvedilol (a non-selective blocker of the β1, β2, and α1 adrenergic receptors) suppressed tumor growth and proliferation [39]. In the low-glucose environment of OSCC, cancer cells can co-opt nociceptive nerves to thrive in nutrient-poor environments [11]. Consequently, drugs that block this process have been proposed to improve the efficacy of nutrient starvation therapy [11]. Further understanding of the nerve–cancer crosstalk in the head and neck TME may provide new nerve-targeted treatment options and help improve clinical outcomes for patients. Accordingly, this review presents the current state of the knowledge on nerve–tumor–immune interactions and the underlying mechanisms, and on neuroscience-driven therapeutic strategies in head and neck cancer (Figure 1). 

## 2. Methods

A comprehensive literature search was conducted using PubMed to identify relevant studies on the topic of nerve–tumor crosstalk in all types of cancer, especially head and neck cancer, published up until December 2023. Combinations of the following keywords were used: head and neck cancer; oral cancer; nerve; peripheral nervous system; tumor neurobiology; perineural invasion; neurogenesis; axonogenesis; neural reprogramming; neuro–immune interactions; and tumor microenvironment. The reference lists of high-impact reviews in the field of interest were also scanned, and articles related to our topic were selected. Priority was given to studies from the last five years, but highly relevant older studies were also included. Data on the study characteristics and outcomes related to mechanisms of nerve–tumor crosstalk were extracted and summarized from the selected articles. The literature search yielded 142 relevant articles on mechanisms of nerve–tumor interactions in head and neck cancer and other solid tumors.

## 3. Perineural Invasion

Perineural invasion (PNI) is a common pathological feature found in cancers of nerve-rich regions, such as prostate cancer, pancreatic cancer, and head and neck cancers [40]. PNI has long been a focus of research because it is considered a route of metastasis in addition to the vascular and lymphatic channels, and multiple studies have associated PNI with an unfavorable prognosis [40,41,42,43]. In this chapter, the main features and mechanisms of PNI are presented, and the limitations in diagnosis are also discussed.

### 3.1. Definition, Diagnostics, and Clinical Relevance

PNI is the extension of malignant tumor cells around, into, or through the nerves and was first reported in head and neck cancer as the tendency to spread along the nerves when migrating into the intracranial fossa [40,44]. The predominant pathological definition of PNI is that the tumor is close to a nerve and involves at least 33% of its circumference, or that tumor cells are found in any of the three layers (the epineurium, perineurium, and endoneurium) of the nerve sheaths [40]. The presence of PNI is often a harbinger of tumor-associated pain and has been correlated with a poor prognosis in several cancers, such as colorectal cancer, prostate cancer, and pancreatic adenocarcinoma [45,46,47]. In HNSCC, particularly OSCC, pretreatment pain is also an important variable that is uniquely associated with PNI [48,49]. In HNSCC, the incidence of PNI varies from 25% to 80% [50,51,52], and several studies have positively related PNI to aggressiveness and decreased survival [23,27,53]. However, a spatial and transcriptomic analysis in OSCC patients not only identified PNI as an independent predictor of a poor prognosis but also suggested that the nerve–tumor distance is an indicator of an unfavorable clinical outcome independent of PNI [23]. 

Adenoid cystic carcinoma (ACC), which accounts for 1% of head and neck cancers and 7.5–10% of salivary malignancies [54], is characterized by PNI, which provides a low-resistance pathway for metastasis and high invasiveness and leads to pain or nerve paralysis at an early stage [55,56]. However, the effect of PNI on ACC remains controversial, with some studies identifying it as a key prognostic factor or independent predictor of local recurrence and metastasis [57,58,59,60], while other studies show no statistical significance for survival [61,62,63]. 

### 3.2. Mode of Mutual Interaction—Schwann Cells 

Schwann cells (SCs) are the major supporting glial cells of the PNS that sheathe the peripheral nerve axons and perform several important functions, including rapid signal transduction, nerve trophic support, extracellular matrix production, neurogenesis, and nerve regeneration [64,65]. SCs can partially dedifferentiate into demyelinated repair SCs when nerves are injured or are invaded by tumor cells [66]. Repair SCs produce neurotrophic molecules and secrete pro-inflammatory factors to remodel the local microenvironment and recruit macrophages to synergistically aid axonogenesis and post-injury repair [66,67]. SCs play a key role in promoting PNI. Deborde et al. demonstrated that SCs enhance cancer invasion through direct interaction with tumor cells [68]. SCs express neural cell adhesion molecule 1 (NCAM1), which separates tumor cell clusters into individual cells to induce their migration towards SCs and spread along nerves. In pancreatic and colorectal cancers, SCs migrate to tumor cells, but not to benign cells, via the NGF-neurotrophic receptor tyrosine kinase 1 (NTRK1/TrkA)–nerve growth factor receptor (NGFR/p75NTR) axis before tumor cells initiate migration towards the peripheral nerves, likely providing a pathway for tumor cell invasion [67,69]. In addition, non-myelinating SCs activated by cancer cells can form tumor-activated Schwann cell tracks (TASTs), which serve as a channel for cancer cell movement and increase cell mobility, similar to the reprogramming of SCs during the nerve repair process [70]. In salivary ACC, SCs have been shown to promote PNI by inducing the epithelial-to-mesenchymal transition (EMT) and the Schwann-like differentiation of tumor cells through the brain-derived neurotrophic factor (BDNF)–neurotrophic receptor tyrosine kinase 2 (NTRK2/TrkB) pathway [71]. SCs are also capable of recruiting immune cells, such as macrophages and cytotoxic T cells, to the PNI site to facilitate tumor neural invasion by generating cytokines [65,67,69]. Taken together, these results show that tumor cells use the normal repair program of SCs to promote PNI.

### 3.3. Limitations and Challenges of PNI Diagnostics

Despite the fact that PNI has a critical impact on tumor prognosis, the accurate diagnosis of PNI is still limited, partly due to the lack of standard diagnostic criteria and the intra- and inter-observer variability. In addition, the limited availability of tumor tissue samples from patients undergoing nonsurgical treatment hinders adequate pathological examination. Therefore, studies have focused on deep learning techniques combined with artificial intelligence and bioinformatic analysis to facilitate PNI diagnosis. Lee et al. developed a deep learning-based human-assisted tool named the Domain-KEY algorithm to help identify PNI in digital slides, which not only promotes diagnostic accuracy but also reduces the diagnostic time [72]. Schmitd et al. performed a spatial transcriptomic analysis of nerves at different distances from the tumor and identified PNI as an independent predictor of a poor prognosis [23]. Furthermore, the study found that the proximity of the nerve to the tumor was also associated with poor outcomes even when the tumor was PNI-negative. These findings suggest that the current diagnostic criteria for PNI should be updated based on the nerve–tumor distance [23]. Weusthof et al. established a PNI-related 44-gene signature based on RNA sequencing data and trained a random forest model to predict occult perineural invasion [73]. However, further efforts are needed to unify the diagnostic criteria for PNI, improve the diagnostic accuracy, and consider the clinical practicality.

## 4. Intra-Tumoral Neural Infiltration

Intra-tumoral nerve infiltration is widely observed in many types of cancer, especially those of highly innervated organs, such as pancreatic cancer, prostate cancer, and head and neck cancer, and it can cause pain, paresthesia, numbness, and paralysis. There is increasing evidence that the intra-tumoral nerve density is associated with tumor progression, metastasis, and prognosis [1,15]. The mechanisms by which tumors regulate intra-tumoral neural infiltration are summarized as axonogenesis, neurogenesis, neural reprogramming, and the transdifferentiation of other cells into neurons (Figure 2). 

### 4.1. Axonogenesis 

Axonogenesis describes neurite sprouting and outgrowth to provide peripheral nerves to the malignant tumor [15]. Ayala et al. demonstrated axonogenesis in human tumors using two- and three-dimensional reconstructions of whole prostates. The study also found that cancer cells induce neurite outgrowth and axon hyperplasia by secreting the axon guidance molecule semaphorin 4F (SEMA4F), and that silencing SEMA4F inhibits experimental neurogenesis [74]. The study provides new evidence that PNI is not the only interaction between tumor cells and nerves. Subsequent studies identified other neurotrophins and axon guidance molecules secreted by cancer cells, such as brain-derived neurotrophic factor (BDNF), nerve growth factor (NGF), and granulocyte-colony stimulating factor (G-CSF), to promote neurite outgrowth and increase the neuronal density in the TME [5,75,76]. Consistently, BDNF is also overexpressed in OSCC and leads to cancer pain via binding to TrkB [77]. NGF, which is highly expressed by OSCC cells, can activate its two receptors TrkA and NGFR, contributing to PNI, cancer pain, and metastasis [78]. 

### 4.2. Neurogenesis

In addition to axonogenesis, Ayala et al. were the first to demonstrate cancer-associated neurogenesis in prostate cancer. While axonogenesis refers to neuron enlargement and axon extension, neurogenesis emphasizes the increase in neuron body cells. The study showed an increase in dorsal root ganglion neurons in prostate cancer patients, suggesting neurogenesis [74]. This phenomenon may also be present in other types of cancers, particularly neurotropic cancers that readily invade in, around, and through peripheral nerves, such as pancreatic and head and neck cancers [79]. More recently, Mauffrey et al. reported a mechanism of cancer-related de novo neurogenesis and demonstrated the migration of doublecortin (DCX)-expressing neural progenitor cells from the subventricular zone (SVZ) of the brain to the tumor site and metastatic niches. At the tumor site, DCX-expressing neural progenitor cells differentiate into adrenergic neurons to promote tumorigenesis [80]. DCX is a microtubule-associated protein that stabilizes microtubules and is involved in neuroblast migration [81]. In a mouse model of prostate cancer, the orthotopic transplantation of DCX+ neural progenitor cells promoted the initiation of both tumor growth and metastasis, whereas the selective depletion of DCX+ progenitor cells suppressed both processes [80]. The study reveals a novel mechanism whereby the SVZ of the CNS is hijacked to provide progenitors for de novo neurogenesis and to facilitate tumor development. However, the underlying mechanism that triggers the recruitment of progenitor cells from the CNS and the molecular signaling pathways involved requires further investigation. The study sheds light on a phenomenon that is easily ignored because of the confusion between the concepts of axonogenesis and neurogenesis, and there is an urgent need to further explore de novo neurogenesis in other cancers.

In addition to being a marker of neural progenitor cells, DCX is expressed in a variety of cancers, including glioblastomas, gangliogliomas, prostate cancer, and liver cancer [81,82,83]. Doublecortin-like kinase 1 (DCLK1), DCLK2, and DCLK3, as members of the DCX superfamily, share common features of appearance and function. DCLK1 has been identified as a cancer stem cell marker in gastrointestinal, pancreatic, and colorectal cancers, and its overexpression has been associated with tumor progression and poor clinical outcomes in several cancer types [84]. DCLK2 and DCLK3 are also associated with tumor invasion and metastasis in breast cancer and gastric cancer, respectively [85,86]. Several studies show DCLK1 overexpression in head and neck cancer, suggesting DCX superfamily-related neurogenesis as a potential mechanism of intra-tumoral neural infiltration [87,88,89].

### 4.3. Neural Reprogramming

To explore the origin of newly formed adrenergic nerves in the TME of head and neck cancer, Amit et al. compared the transcriptome of tumor-associated trigeminal sensory neurons with that of endogenous neurons and revealed a phenotypic switch from sensory nerves to adrenergic neo-neurons induced by cancer-derived extracellular vesicles (EVs) [39]. This phenomenon has been linked to *TP53*, one of the most frequently mutated tumor suppressor genes in head and neck cancer, and its fluctuating expression is also closely associated with nerve regeneration [90,91]. In *TP53*-deficient tumors, a miRNA array analysis revealed a decrease in miR-34a and miR-141 in EVs, which caused an increase in the number of neurofilaments and promoted the axonogenesis of trigeminal ganglion neurons. The reduction in miR-34a also induced the transdifferentiation of sensory nerves towards adrenergic nerves [39]. By performing surgical sensory denervation and chemical sympathectomy prior to tumor cell inoculation in a mouse model, they showed that exosome-induced neural reprogramming, rather than the outgrowth of existing adrenergic nerves, promoted tumor proliferation and progression [39]. In conclusion, the study identified a novel mechanism of nerve–cancer interaction in head and neck cancer and proposed a potential strategy for anti-cancer therapy.

### 4.4. CSC Differentiation into Neurons

Cancer stem cells (CSCs) are a group of undifferentiated tumor cells that possess self-renewal, multipotency, and differentiation capabilities. CSCs play a critical role in driving tumorigenesis, metastasis, and recurrence [92,93]. One study demonstrates that the neurotransmitter 5-hydroxytryptamine (5-HT) produced by enteric serotonergic neurons is able to modulate the self-renewal and tumor-initiation capacities of colorectal CSCs [94]. Lu et al. observed that neural cells with human cell-specific markers were detectable in xenografts after the transplantation of CSCs from gastric and colorectal cancer patients into nude mice. In an in vitro differentiation assay, they provided experimental evidence that a fraction of CSCs has the capacity to generate neuronal cells. These CSC-derived neural cells express tyrosine hydroxylase (TH) and vesicular acetylcholine transporter (VaChT), the neuronal markers of sympathetic and parasympathetic neurons, respectively [95]. The notion that CSCs serve as a source of tumor neurogenesis was further supported via RNA sequencing analyses of aldehyde dehydrogenase (ALDH)-positive CSCs from colon cancer patient-derived organoids (PDOs) and xenografts (PDXs). The study confirmed an enrichment of neural developmental gene expression in CSCs. Moreover, the functional analyses demonstrated a key role for the neural crest stem cell (NCSC) regulator early growth response 2 (EGR2) in tumor growth, suggesting that targeting EGR2 may provide a therapeutic differentiation strategy to eliminate CSCs and block nervous system-driven disease progression [96]. In ACC, a study revealed a previously uncharacterized CSC population with neural stem cell (NSC) properties expressing SRY-box transcription factor 10 (SOX10), a marker of the gliogenesis and maintenance of adult NSCs and other neural differentiation factors, such as notch receptor 1 (NOTCH1) and fatty acid binding protein 7 (FABP7) [97]. A subsequent study also found that SOX10-expressing and NSC-like CSCs in basal-like breast carcinoma shared characteristics and provided new targets for treatment [98]. 

### 4.5. Macrophage Transdifferentiation into Neurons

In addition to neuronal cells, cancer pain is also associated with numerous non-neuronal cells mainly through the secretion of pro-inflammatory mediators or algogenic mediators to sensitize nociceptors in the TME [99]. Monocytes and macrophages can produce tumor necrosis factor (TNF) and interleukin 1 beta (IL-1β) to enhance pain transduction and conduction, and the depletion of these cells conversely impairs the development of mechanical and thermal hypersensitivity [100,101]. Taken together, these results suggest that tumor-associated macrophages (TAMs), as an important component of the TME, have the potential to modulate intra-tumoral neural infiltration. Based on this hypothesis, Tang et al. used the single-cell RNA sequencing of lung adenocarcinoma to uncover a macrophage-to-neuron-like cell transition (MNT), describing a phenomenon in which TAMs directly transdifferentiate into neuron-like cells to facilitate de novo neurogenesis [102]. In vitro, bone marrow-derived MNT cells exhibited neuronal phenotypes and activities, and NOD/SCID mice showed increased tumor-related nociceptive behavior following MNT transfer [102]. SMAD family member 3 (SMAD3), a modulator of the neural lineage differentiation of pluripotent NSCs, served as a key regulator for the genetic promotion of the MNT, and its blockage reduced tumor innervation and tumor-related nociceptive behaviors in vivo [102,103]. Overall, the study provides a novel mechanism of neurogenesis that may represent a precision therapeutic target for cancer pain and tumor progression.

## 5. Neuro–Immune Interactions

Mutual interactions between the nervous and immune systems play a critical role in maintaining homeostasis. In the context of cancer, neurons and immune and tumor cells interact via soluble signaling molecules and neurotransmitters to form a triangular relationship that modulates tumor growth and progression. However, sensory nerves, autonomic nerves, and other neural components have diverse effects on the regulation of the immune phenotype and the anti-tumor immune response [1,104] (Figure 3).

### 5.1. Sensory Nerves

Nociceptors are the most widely studied sensory nerve fibers in the context of the immune surveillance and immune escape of cancers. Nociceptors can influence immune components by releasing neuropeptides, such as CGRP, and, in melanoma, CGRP released by nociceptors directly promotes CD8+ T cell exhaustion [104,105]. Indeed, the pharmacological nociceptor inhibition or blocking of the receptor activity modifying protein 1 (RAMP1), the receptor for CGRP, reduced T cell exhaustion and inhibited tumor growth in a mouse model. CD8+ T cell exhaustion was restored in a mouse model in which most mechano- and thermosensitive nociceptors were ablated via treatment with recombinant CGRP. Analysis of single-cell RNA sequencing data from human melanoma samples showed that CD8+ T cells expressing RAMP1 are more likely to be exhausted than RAMP1-negative cells and are associated with a poor prognosis [106]. CGRP also suppresses the murine macrophage antigen presentation to T cells by modulating the cytokine expression and regulating macrophage polarization to the pro-tumorigenic M2 subtype [107].

In the head and neck region, tissues are innervated by sensory nerves, primarily originating from the trigeminal ganglia [39,108]. CGRP, as the most abundant neurotransmitter in trigeminal ganglion neurons [108], has been shown to regulate the immune response through the RAMP1 signaling pathway in OSCC [109]. This study demonstrated a higher expression of RAMP1 for tumor-infiltrating immune cells (CD4+ T cells, cytotoxic CD8+ T cells, and NK cells) in OSCC patients compared to immune cells from the normal tissue of healthy individuals [109]. High expression of RAMP1 was also confirmed in cultured oral cancer cells and orthotopic xenografts, contributing to oral cancer pain [110]. In *Cgrp* knock-out mice, tumor growth was significantly inhibited compared to wildtype controls, accompanied by an increase in tumor-infiltrating immune cells [109]. 

Another study identified and functionally tested a sensory neuro-immune circuit that responds to lymph-borne inflammatory signals. Transcriptomic profiling revealed that multiple sensory neuron subsets, predominantly peptidergic nociceptors, innervate lymph nodes (LNs). The optogenetic stimulation of LN-innervating sensory fibers induced rapid transcriptional changes in several cell types, and particularly in endothelia, stromal cells, and innate leukocytes [111]. 

### 5.2. Autonomic Peripheral Nerves

Stress can promote tumor initiation and progression in several cancer types, including oral, prostate, and skin cancers [112,113]. This is explained, in part, by the fact that the stress-induced adrenergic signaling produced by local sympathetic nerves has multiple effects on different immune components of the TME. Accordingly, stress has been linked with increased metastasis in breast cancer, in part by affecting the recruitment of tumor-associated macrophages (TAMs), myeloid-derived suppressor cells (MDSCs), and other immune cells [114]. 

Stress-induced β-adrenergic receptor (β-AR) signaling causes DNA damage, leading to tumorigenic effects in human oral keratinocytes [115]. In ovarian cancer cells, exposure to norepinephrine (NE) causes DNA double-strand breaks, and pretreatment with propranolol (a non-selective β-blocker) can counteract the norepinephrine-induced DNA damage [116]. Similarly, NE significantly impairs the DNA damage repair capacity of UV-exposed murine NIH/3T3 fibroblasts via the regulation of the DNA damage sensors checkpoint kinase 1 (Chek1/Chk1) and Chk2, and the proto-oncogene cell division cycle 25A (Cdc25A), which is involved in cell cycle delay [117].

Other studies have reported that the immunomodulatory cytokines IL-6 and IL-8, which are important in inflammation and tumor development, are induced via β-AR signaling to enhance the growth and prevent the apoptosis of tumor cells via the SRC proto-oncogene, non-receptor tyrosine kinase (SRC), or cAMP/PKA signaling pathways [118,119,120]. In addition, Guillermo et al. demonstrated a significant increase in C-C motif chemokine ligand 2 (CCL2/MCP1) via cAMP and PKA after the stimulation of β-AR signaling, resulting in macrophage recruitment and infiltration to tumor sites [121]. Erica et al. showed a 30-fold increase in breast cancer metastasis via β-AR signaling via the promotion of the infiltration of macrophages and their differentiation into the M2 subtype [114].

Immune checkpoint molecules, such as programmed death-1 (PD-1) and programmed cell death ligand-1 (PD-L1), are critical for immune surveillance and anti-tumor immune responses. It is worth noting that norepinephrine activity via β_2_AR signaling can upregulate PD-1 on T cells [122]. A study in prostate cancer showed a high expression of PD-L1 in regions with abundant intra-tumoral nerves, and that the density of PD-L1-positive tumor-associated nerves is negatively correlated with the quantity of CD8+ T cells [123]. In breast cancer, sympathetic nerve stimulation accelerates tumor growth while parasympathetic nerve stimulation has the opposite effect. Tumor-specific sympathetic denervation in a mouse xenograft model and in a rat model with chemically induced tumors showed a marked decrease in the PD-1 and PD-L1 expressions on CD8+ and CD4+ T lymphocytes [6].

### 5.3. Other Neural Components

Schwann cells (SCs) are another notable member of the peripheral nervous system that can produce chemokines to enhance the chemotactic capacity of immune cells [124]. SCs modulate the immune microenvironment through the CCL2/ C-C motif chemokine receptor 2 (CCR2) axis. CCL2 released by SCs enhances the proliferation, migration, and invasion capabilities and EMTs of tumor cells, as well as induces TAMs to polarize into the M2 subtype, resulting in immunosuppression [125,126]. Toll-like receptors (TLRs), the activation of which modulates the maturation of antigen-presenting cells and T cell activation, are also expressed on SCs [104,127,128].

However, neural signaling molecules can sometimes exert functions directly on immune cells without involving neural structures. The neurotransmitter gamma-aminobutyric acid type A (GABA), produced and secreted by B lymphocytes, can promote the differentiation of monocytes into anti-inflammatory macrophages that release IL-10. Binding to the corresponding receptor on CD8+ T cells inhibits the anti-tumor response and promotes tumor growth [129]. In orthotopic xenograft mouse models of gastric and pancreatic cancer, platelet-derived serotonin upregulated the PD-L1 expression in tumor cells via histone serotonylation mediated by transglutaminase 2, impaired the function of intra-tumoral CD8+ T cells, and accelerated tumor growth [130].

## 6. Imaging and Treatment Strategy

The study of nerve–cancer interactions relies on the development of imaging techniques and the establishment of in vitro and in vivo models. The rapid development of multi-omics analyses is also providing novel approaches to investigate the potential mechanisms at the molecular level. These research tools not only help to elucidate the nature of tumor neurobiology but also facilitate the clinical translation of nerve-specific treatment strategies. 

### 6.1. Imaging Technologies for Cancer–Neuron Interactions 

To better understand the interactions between the PNS and tumor, multidisciplinary techniques and methods are required to provide clear insights from the morphological to the molecular level. At the structural and functional level, electron microscopy (EM) is widely used to reveal the synaptic structures between neurons and tumor cells [131]. Dynamic contrast-enhanced magnetic resonance (MR) and MR spectroscopy are utilized for the neuroimaging of in vivo models or clinical diagnosis [132,133]. In CNS tumors, multiphoton laser-scanning microscopy (MPLSM), which provides high-resolution in vivo imaging, has been used to monitor the coordinated activity of tumor cells in neuronal signaling when combined with calcium imaging in a spatial and functional manner [131,134,135]. Imaging techniques for non-CNS tumors are less developed because of the limitations of tracking the entire peripheral nerve system. To improve this situation, tissue-clearing techniques and light-sheet microscopy have been developed to achieve the visualization of relevant structures at the subcellular level through intact, transparent organs, facilitating the assessment of physical and functional cancer–neuron interactions [14,136,137]. However, more emerging neuroimaging techniques should be leveraged to map nerve involvement and interactions longitudinally in vivo. At the molecular level, multi-omics analyses, such as single-cell RNA sequencing, spatial transcriptomics, and proteomic approaches, have been developed to investigate the molecular characteristics of neurons, tumor cells, and other cellular components in the TME [1,23,73]. Other techniques to explore the mechanism underlying the cancer–neuron crosstalk include the generation of enhancer-based lentiviruses, single-molecule imaging tools, and in vivo genetic perturbation [1]. In addition, in vitro co-culture models and in vivo xenograft mouse models are often used to better understand the dynamic interactions between the neurons, tumor cells, and components of the TME [1].

### 6.2. Neuroscience-Instructed Therapeutic Approaches 

The emerging clinical relevance of the reciprocal cancer–neuron interactions within the TMEs of many human tumors has attracted great interest in the development of innovative therapeutic strategies, either based on the identification of new drug targets or the repurposing of already well-established drugs for anti-tumor treatment. For example, β-blockers are widely used as drugs to regulate blood pressure, heart rate, and airway reactivity [138], but they also show anti-tumor effects due to their antagonistic actions on the adrenergic nervous system [39,139,140]. In breast cancer, a phase II clinical trial demonstrated that propranolol decreased the expressions of biomarkers, which are associated with EMT-related signaling and metastasis in treated patients [141]. Other clinical trials showed tumor suppression in melanoma, prostate cancer, and hepatocellular carcinoma, via the targeting of adrenergic receptors [14]. These findings suggest that β-blockers may work synergistically with other well-established anti-tumor drugs to improve the clinical outcomes of cancer patients [138]. Although several neural signaling inhibitors have shown promising anti-tumor efficacy in vitro and in vivo, challenges remain in the clinical application not only because these drugs are not TME-specific, but also because they affect the normal physiological processes and function of the PNS. 

Local denervation via surgical or pharmacological intervention is another potential strategy to treat cancer patients, especially those with neurotropic carcinomas. Microsurgical denervation, which minimizes the extent of surgical intervention, is considered an alternative for cancer patients with a high risk of undergoing direct tumor resection [14]. In prostate cancer, a phase I/II clinical trial performing denervation via unilateral botulinum toxin type A injection prior to prostatectomy showed increased tumor cell apoptosis compared to contralateral tumor tissue [142]. However, the strategy still faces the challenge that denervation has potential side effects on the organ system.

In head and neck cancer, the translation of PNS-targeted therapy from preclinical models to clinical practice is limited, in part because the abundant and complex nerve distribution in this anatomic region restricts the adequate inhibition of neurons. The development of new targeted therapies based on neuron-specific strategies requires a better understanding of the underlying principles of the complex cancer–neuron crosstalk.

## 7. Conclusions and Outlook

In summary, recent studies have demonstrated that the nervous system can directly or indirectly regulate tumor progression and metastasis, while tumors actively recruit neural components to remodel the TME. The study of the nerve–tumor interactions in several cancer types, including head and neck cancer, has elucidated several potential mechanisms that facilitate the development of nerve-targeted treatment strategies. In head and neck cancer, nerve–tumor crosstalk plays a critical role in tumor growth and contributes to unfavorable clinical outcomes. The establishment of deep learning techniques and bioinformatic approaches has not only improved the early diagnosis of PNI but also paved the way for innovative and comprehensive treatment options. The understanding of intra-tumoral neural infiltration in head and neck cancer also provides potential strategies targeting denervation. 

In terms of treatment, the current nerve-targeted strategies lack specificity for neural components of the TME and therefore have limited use in clinical practice due to side effects on normal nerve function. It is anticipated that a more detailed characterization of nerve and tumor cell signaling and biomarkers at the molecular level will help identify potential druggable targets, leading to the development of neuroscience-based TME-targeted treatment options. Advances in drug delivery systems to locally target nerve–tumor signaling pathways may also facilitate overcoming these challenges.

While this review synthesizes the existing evidence on nerve–tumor interactions, the studies conducted to date have some limitations, such as unclear clinical relevance and modes of action for different head and neck cancer subtypes, small sample sizes, or a lack of experimental validation in clinical samples. The future direction of this field should focus on elucidating specific signaling pathways involved in processes such as perineural invasion and neural reprogramming in the head and neck cancer subtypes. In vivo models that can better simulate the complex nerve–tumor microenvironment are also urgently needed to elucidate the mutual interactions between tumors and the nervous system, which is critical for the future translation into clinical practice. Integrating basic knowledge of neuro-immune signaling will also help to establish combination therapies in order to disrupt lethal nerve–tumor communication.

## Figures and Tables

**Figure 1 cells-13-00256-f001:**
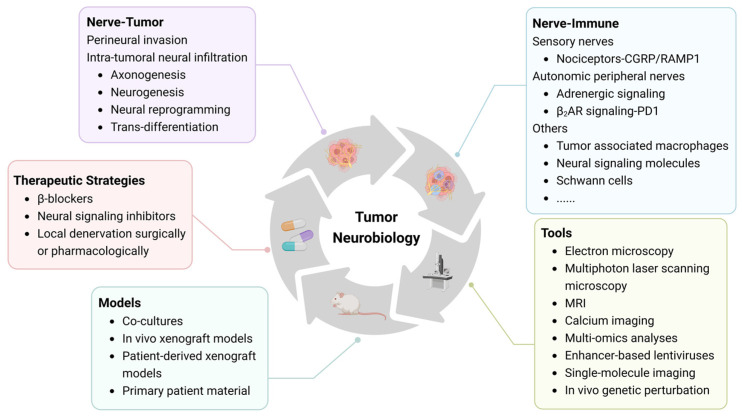
Summary of nerve–tumor–immune interactions, research tools, and neuroscience-based therapeutic strategies discussed in this review article.

**Figure 2 cells-13-00256-f002:**
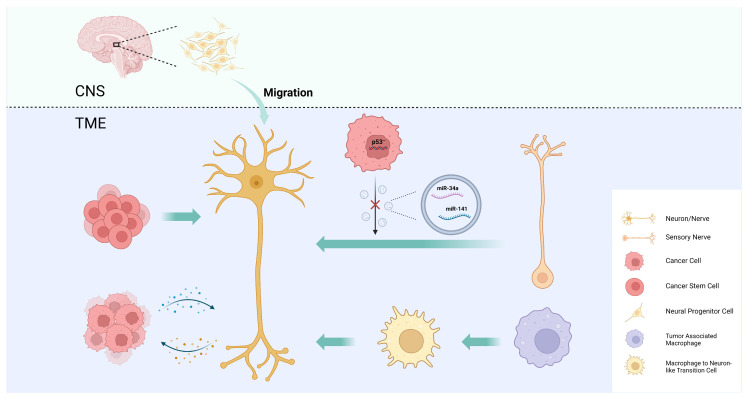
Mechanisms of intra-tumoral neural infiltration. Tumors regulate intra-tumoral neural infiltration through axonogenesis, neurogenesis, neural reprogramming, and the transdifferentiation of other cells into neurons.

**Figure 3 cells-13-00256-f003:**
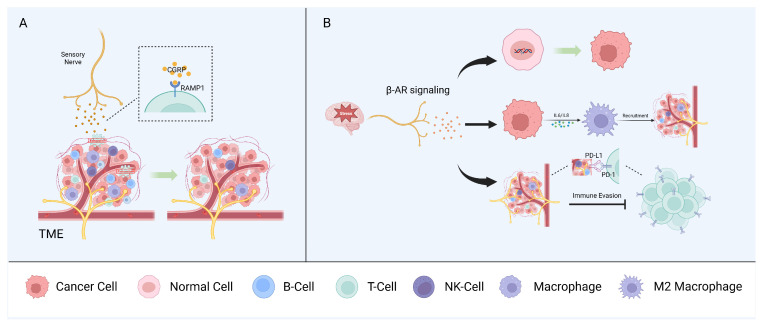
Neuron–immune interactions. Sensory nerves (**A**) and autonomic peripheral nerves (**B**) interact with immune components of the TME and promote tumor growth synergistically.

## Data Availability

No new data were created or analyzed in this study. Data sharing is not applicable to this article.

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
