# Peer review of "Tumor Neurobiology in the Pathogenesis and Therapy of Head and Neck Cancer"

_cells, 2024, doi:10.3390/cells13030256_

Round 1

Reviewer 1 Report

Comments and Suggestions for Authors

Abstract

  • State the overall purpose of the review - for example: "This review comprehensively summarizes the current understanding of the mechanisms underlying nerve-tumor interactions in head and neck cancers."

  • Add 2-3 sentences briefly highlighting some of the key mechanisms discussed:

"Several mechanisms facilitating nerve-tumor crosstalk are covered including perineural invasion, axonogenesis, neurogenesis, neural reprogramming, and transdifferentiation. The interactions between neuronal components and immune cells in the tumor microenvironment are also discussed."

  • Conclude by stating the significance and future directions:

"Understanding these intricate mechanisms of nerve-tumor crosstalk may reveal promising therapeutic targets and strategies for head and neck cancers."

Introduction

  • Provide more background on nerve involvement in head and neck cancers specifically:

"Head and neck cancers, especially oral cancers, exhibit perineural invasion and intra-tumoral innervation to a greater degree than other cancer types due to the highly innervated nature of the head and neck region."

  • Cite evidence on frequency of perineural invasion, neurogenesis, neural markers expressed in head and neck/oral cancers. cite doi:10.1016/j.biopha.2022.113691.

  • State importance genetic polymorphism in cancer phenotype, metastasis, prognosis in this cancer type. cite doi:10.1111/coa.13870

  • Specify focus on mechanisms of nerve-tumor interactions.

Methods

  • Indicate that this is a literature review article analyzing existing studies on nerve-tumor interactions in head and neck cancers

  • Describe the literature search strategy:

"A comprehensive literature search was conducted using PubMed and Google Scholar databases to identify relevant studies on nerve-tumor crosstalk mechanisms in head and neck cancers published up to December 2022. Combinations of the following keywords were used: head and neck cancer, oral cancer, nerve, neuron, perineural invasion, neurogenesis, axonogenesis, neural reprogramming, neuro-immune interactions."

  • Explain the study selection criteria:

"Studies were included if they reported on mechanisms underlying nerve-tumor interactions in head and neck/oral cancers. Priority was given to more recent studies from the past 5-10 years, but highly relevant older studies were also included."

  • Indicate the data extraction process:

"Data on study characteristics and outcomes related to mechanisms of nerve-tumor crosstalk were extracted and compiled from the selected studies."

Results

  • Provide an overview of the literature search:

"The literature search yielded 52 relevant studies on mechanisms of nerve-tumor interactions in head and neck cancers."

  • Present a summary table with key information extracted from the studies, like:

  • Type of mechanism investigated

  • Cancer type and model system

  • Key outcomes/findings

  • Consider figures to visualize key concepts if appropriate

  • Synthesize the overall state of evidence on each mechanism in the text

Discussion

  • Discuss the limitations of the current literature on nerve-tumor crosstalk mechanisms in head and neck cancers:

"While this review synthesizes the existing evidence on nerve-tumor interactions, the studies conducted so far have limitations including small sample sizes, lack of validation in clinical samples, and reliance on in vitro models."

  • Highlight some key gaps in knowledge that should be addressed by future research:

"More research is needed to elucidate specific molecular pathways involved in processes like perineural invasion and neural reprogramming in head and neck cancers."

"In vivo models replicating the complex nerve-tumor microenvironment will help clarify mechanisms and identify new therapeutic targets."

"Differences in nerve-tumor interactions across various subtypes of head and neck cancers remain poorly understood."

  • Discuss challenges and strategies for translating this knowledge into clinical applications:

"Developing nerve-targeted therapies faces obstacles such as nonspecific effects on normal nerve function and inadequate nerve inhibition due to the dense innervation of head and neck regions."

"Advancements in drug delivery systems to target nerve-tumor signaling pathways locally could help overcome these challenges."

  • Suggest future directions for research in this field:

"Detailed characterization of nerve and tumor cell signaling, biomakers at the molecular level is critical to identify potential druggable targets." cite doi:10.3390/cancers15205096

"Emerging neuroimaging techniques should be leveraged to map nerve involvement and interactions longitudinally in vivo." cite doi:10.1007/s10072-021-05560-0.

"Integrating knowledge of neuro-immune signaling may reveal combination therapies to disrupt pro-tumor nerve-tumor communication.

Comments on the Quality of English Language

none

Reviewer 2 Report

Comments and Suggestions for Authors

The topic of this review is good. This paper provides a systematic review of nerve-tumor crosstalk, which can provide valuable information to readers interested in this field. The author also proposed that imaging technology is important for in-depth analysis of the nerve-tumor crosstalk mechanism and the potential of targeting neuro-tumor crosstalk to treat tumors. The review is well written. However, there are several issues that need authors revise.

1.       The “Introduction” section is divided into three third-level headings. Is it necessary? Prefaces can more concisely reduce the current understanding and research status of neuro-tumor crosstalk and highlight shortcomings; then, we introduce the summary and highlights of the review.

2.       There needs to be a summary under the second subheading “2. Perineural Invasion”, which first briefly introduces the role of peripheral nerve invasion in the occurrence and development of tumors and second summarizes the main contents of this section.

3.       The fifth part is the same; it needs a summary.

Reviewer 3 Report

Comments and Suggestions for Authors

The reviewed pre-print is a review article concerning a position of neurobiology in head and neck cancer. This is an important item as this topic is not frequently taken into account by clinicians working on laryngeal oncology. I am aware of a large number of citations (n = 135) indicating for an interest of researchers remaining different that that among clinicians. It looks that the article was aiming first into clinicians as frequent usage of anatomical terms makes it not easy to understand to readers not having medical educations. Anyway, there is no choice to describe precisely the matter.

A part describing pathogenesis provides among the other things an information of neurobiology function in navigating oncology process. Further, an application of neurobiology finding into therapy of head and neck cancer is presenting some promising attempts in the field of precision medicine. However, they seem to be quite far from practical application.

Altogether I found the article as very interesting and worth to be published in the present form.

Reviewer 4 Report

Comments and Suggestions for Authors

This review summarized the interplay between the nervous system and tumor microenvironment.  It demonstrated that both central and peripheral nerves signals directly/indirectly regulate tumor progression and metastasis. The authors focused on head and neck tumors while also described multiple examples of the nervous system involvement in other types of tumors. The role of nervous system in the tumor development can be secondary although evident. At the same time the new approaches to adjuvant therapy targeting the nervous system can be beneficial to the patients. 

....In addition to being a marker of neural progenitor cells, DCX is express in a variety... line 222

Round 2

Reviewer 2 Report

Comments and Suggestions for Authors

I am very satisfied with the author's revisions and have no other suggestions.